# *Citrobacter rodentium* infection activates colonic lamina propria group 2 innate lymphoid cells

Rita Berkachy⊚, Vishwas Mishra⊚, Priyanka Biswas, Gad Frankel ⊚ *

Department of Life Sciences, Imperial College London, London, United Kingdom

⊚ These authors contributed equally to this work.
* g.frankel@imperial.ac.uk

## Abstract

Group 3 innate lymphoid cells (ILC3s) play a major role in protecting against infection with the enteric mouse pathogen *Citrobacter rodentium* (CR) used to model infections with enteropathogenic and enterohaemorrhagic *Escherichia coli*. ILC3s-secreted IL-22 induces secretion of IL-18, antimicrobial peptides and nutritional immunity proteins as well as activation of tissue regeneration processes. While ILC2s have traditionally been associated with immune responses to helminth infection and allergic inflammation via the production of type 2 cytokines (e.g. IL-4, IL-5, IL-9 and IL-13), more recently they have been implicated in protection against *Clostridium difficile* and *Helicobacter pylori* infections. Here we show that colonic lamina propria ILC2s expand in response to CR infection and secrete IL-4, IL-5 and IL- 13, which are involved in maintenance of the intestinal barrier function, tissue repair and mucus secretion. When stimulated with IL-18, and IL-33 as a control, colonic ILC2s from uninfected mice secreted type 2 cytokines. Injection of IL-18 binding protein (IL18 BP), at 2- and 3-days post CR infection, blocked expansion of ILC2s in vivo. While ILC2s do not expand in CR-infected *Il22*-/- mice, injection of IL-18 into *Il22*-/- mice at 2- and 3-days post CR infection triggered ILC2s expansion. Importantly, injection of anti-IL-13, at 2- and 4-days post CR infection, diminished local secretion of IL-10 and IL-22. These data show that ILC2s are activated in response to infection with an enteric Gram-negative pathogen. Moreover, stimulation with IL-18 plays a role in ILC2s expansion and secretion of type 2 cytokines, which may participate in shaping the local immunological landscape.

## Author summary

While group 3 innate lymphoid cells (ILC3s) play a key role in protection against bacterial infections, ILC2s are mainly associated with immune responses to helminth infections. Here we show that infection with the enteric mouse pathogen *Citrobacter rodentium* (CR) induces gut ILC2s expansion and secretion of

**Data availability statement:** All the data generated in this study are available here: https://doi.org/10.6084/m9.figshare.29127863.v1.

**Funding:** This study was supported by grants from The Wellcome Trust (https://wellcome.org) (107057/z/15/z and 224282/Z/21/Z GF) and the Medical Research Council (MRC) (https://www.ukri.org/councils/mrc/) (MR/R02671to GF). The funders had no role in study design, data collection and analysis, decision to publish, or preparation of the manuscript.

**Competing interests:** The authors have declared that no competing interests exist.

type-2 cytokines. ILC2 isolated from uninfected mice were activated by IL-18. Consistently, administration of IL-18 binding protein into CR-infected mice inhibited ILC2 expansion. Conversely, injection of IL-18 into CR-infected *Il22*-/- mice induced ILC2 expansion. These findings suggest that gut ILC2s are activated by Gram negative enteric pathogens, which is mediated in part by IL-18.

## Introduction

*Citrobacter rodentium* (CR) is the etiologic agent of transmissible murine colonic crypt hyperplasia (CCH) [1–3]. It is an enteric extracellular Gram-negative pathogen, which serves to model infections with enteropathogenic *Escherichia coli* (EPEC) and enterohemorrhagic *E. coli* (EHEC) [4,5]. To bind intestinal epithelial cells (IECs) and proliferate, CR employs a filamentous type III secretion system (T3SS), which injects effector proteins directly into IECs [6]. Following injection, the effectors form an intracellular network which subverts signal transduction and cellular functions in the host cell, including trafficking, tight junctions, cell death, NF- kB, JNK/p38 and caspase-4/11, -8, and -9 [7–9]. Infection of CR (as EPEC and EHEC) is unique, as the pathogen injects its own receptor, Tir, into infected IECs [10]. Binding of the outer membrane adhesin intimin to Tir leads to intimate bacterial attachment, effacement of the brush border microvilli, induction of localised actin polymerisation and formation of a pedestal-like structure under the attached bacteria [11].

The CR infection cycle is divided into four phases [12,13]: An establishment phase (1–3 days post infection (dpi)), where the pathogen colonises the caecal lymphoid patch [14]. An expansion phase (4–8 dpi), initially characterised by activation of group 3 innate lymphoid cells (ILC3s)-secreting IL-22 (4 dpi), which is believed to be essential for survival [15–18]. Binding of IL-22 to the IL-22 receptor in IECs activates STAT3 and triggers expression of IL-18, antimicrobial peptides (e.g. Reg3β and Reg3γ) and nutritional immunity (e.g. LCN-2 and calprotectin) promoting epithelial barrier resistance [18–20]. This is followed (6–8 dpi) by rapid CR proliferation, adherence to IECs in the distal colon and development of CCH [13]. A steady- state phase (9–12 dpi), characterised by high and stable CR shedding and a switched of IL-22 production from ILC3s to CD4+ T cells, which induces sustained STAT3 activation in both superficial and crypt IECs, limiting bacterial dissemination [18,21]. A clearance phase (from 12 dpi), when CR is rapidly cleared via opsonisation with IgG and phagocytosis by neutrophils [21,22].

ILC2s have been implicated in immune responses to extracellular helminth infections via the secretion of the type 2 cytokines IL-4, IL-5, IL-9 and IL-13 [23,24]. However, recent studies have shown that secretion of IL-5 and IL-13 from ILC2s protects mice against toxin-mediated epithelial damage during *Clostridium difficile* infection (CDI) [25,26]. Administration of IL-13 protected mice from CDI-induced weight loss and severe disease, while anti-IL-13 exacerbated infection outcomes [27]. During *Helicobacter pylori* infection, ILC2-secreting IL- 5 influenced B cell responses and enhanced IgA coating of the bacteria, helping control the infections [28].

ILC2s are mainly activated by IL-25, IL-33 and TSLP which are released from epithelial and stromal cells [29]. More recently, IL-18 has been shown to activate resident ILC2s in the skin [30]. IL-33 is expressed by IECs in CR-infected mice and higher CR faecal colony forming units (CFUs) were recovered from $Il33^{-/-}$ compared with wild type (WT) mice at 14 dpi [31]. In contract, $Il25^{-/-}$ presented only subtle difference in CR infection outcomes compared to WT mice [32]. The role of TSLP in CR infection is not known. IL-18 has been shown to have a protective role following CR infection, as $Il18^{-/-}$ mice presented greater bacterial load and exacerbated histopathology [33,34].

While IL-33 and IL-18 have been shown to play a role in CR infection, it is not known if they signal to colonic lamina propria ILC2s. In this study we found that the ILC2 population expanded upon CR infection at 4 dpi, when ILC3s are also activated [15,16,35]. ILC2s isolated from CR-infected mice produced elevated amounts of IL-5, IL-4, and IL-13. Stimulation of ILC2s, isolated from the colons of naïve mice with IL-18, and IL- 33 as a control, resulted in proliferation and secretion of IL-5, IL-9, and IL-13. While administration of IL-18 BP into CR-infected WT mice inhibited ILC2s expansion, injection of recombinant IL-18 into CR-infected $Il22^{-/-}$ mice restored ILC2 expansion. These findings suggest that in parallel to ILC3s, ILC2s are also activated during the expansion phase of CR infection.

## Results

### Colonic ILC2s expand in response to CR infection

The aim of this study was to determine whether ILC2 respond to enteric Gram-negative bacterial infections. First, we infected C57BL/6 mice with CR by oral gavage and enumerated faecal CFU at 4 dpi, when it is known that ILC3s-derived IL-22 activates IECs [18]. This revealed shedding at ca. $10^8$ CFU/g of faeces (Fig 1A). We then measured crypt length in H&E-stained paraffin-embedded colonic sections, which showed no significant increase in CCH compared to the uninfected control mice (Fig 1B and 1C). In contrast, the colon length of CR-infected mice was significantly shorter than PBS-dosed mice and faecal myeloperoxidase (MPO), a neutrophil-derived biomarker, was significantly higher than the control (Fig 1D-F). Once we confirmed that the mice were infected and responded to the pathogen, we investigated colonic lamina propria ILC2s; mice dosed with PBS were used as controls. We assessed the numbers and proportions of ILC2s by flow cytometry, using a combination of lineage markers, CD45, KLRG1, CD127 and the transcription factor GATA-3 (Fig 1G). This revealed that the number and proportion of ILC2s increased significantly in the CR-infected mice (Fig 1H). We also observed an increased proportion of NK cells following infection, with no significant changes in the frequencies of ILC1 or ILC3 populations (S1A Fig). Consistent with previous reports [15], ILC3s were activated, as evidenced by elevated levels of IL-22 in colonic explants and increased faecal lipocalin-2 levels (S1B and S1C Figs).

### CR infection induces type 2 cytokines secretion

To determine their cytokine profile, we sorted ILC2s from pooled colons of uninfected and CR-infected mice at 4 dpi (Fig 2A). Upon stimulation with PMA/ionomycin (without protein transport inhibitors), the levels of the type 2 cytokines were measured by ELISA. This revealed that ILC2s isolated from CR-infected mice secreted elevated amounts of IL-4, IL-5, and IL-13 (Fig 2B). When protein transport inhibitor cocktail was present in the stimulation media, IL-4$^+$, IL-5$^+$ and IL-13$^+$ ILC2 populations were higher in CR infected samples (Fig 2C and 2D), suggesting that colonic lamina propria ILC2s not only expanded but were also activated in CR-infected mice.

### Colonic ILC2s proliferate in response to IL-18 ex vivo

We next evaluated if any of the ILC2-activating cytokines are expressed in colon 4 days post CR infection. We extracted mRNA from colonic tissue and performed RT-qPCR. This revealed that while $Il25$ expression did not increase and $TLSP$ was not detected, expression of $Il18$ and $Il33$ was elevated in CR-infected mice (Fig 3A). Moreover, ELISA revealed that IL-18 levels, secreted from tissue explants, were significantly higher in the infected samples (Fig 3B).

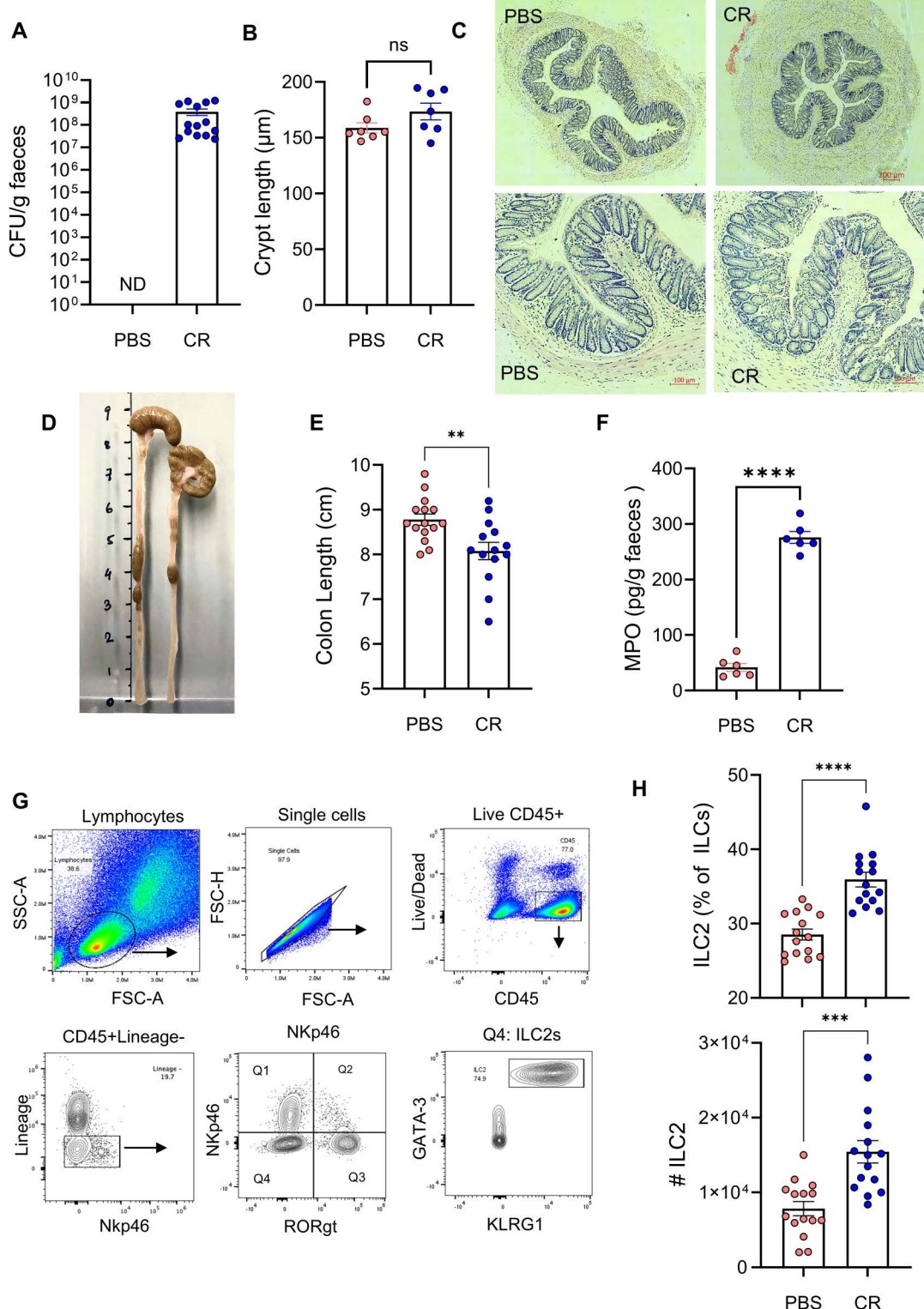

**Fig 1. Colonic lamina propria ILC2s expand in response to CR at 4 dpi.** A. Faecal CR (CR) at 4 dpi. Mock infected (PBS) were used as controls. B. Crypt lengths measured in H&E-stained colon sections. C. Representative H&E sections from the distal colons of uninfected (PBS) and infected mice at 4 dpi (scale bar 100 μm). D. Representative and E. quantitative colon length. F. Faecal MPO levels measured by ELISA. G. Flow cytometry gating

strategy used to identify ILC2s from mice colonic lamina propria cells. Debris (SSC-A vs. FSC-A) and doublets (FSC-H vs. FSC-A) were excluded, and live/dead discrimination was determined using the LIVE/DEAD fixable blue dead cell stain kit (FSC-A vs. Live/Dead). Total ILCs were defined as CD45+NKp46+/-Lineage- [CD3/B220/CD19/TER119/Gr-1/CD5/CD11c/CD4/Nk1.1]. Expression of the CD127, KLRG1 and GATA3 markers were used to identify ILC2 (CD127+KLRG1+GATA3+). H. Numbers and frequency of ILC2s increased upon infection. dpi: days post infection. Data shown are pooled values from three independent experiments. P values were determined on data plotted as mean±SEM using Student's-t-test. ns: non-significant; **p<0.01; ***p<0.001; ****p<0.0001.

We investigated if, similarly to skin ILC2s, IL-18 could also stimulate colonic lamina propria ILC2s [30]. To this end, ILC2s sorted from the colons of naïve mice were cultured for 3 days in complete RPMI medium containing recombinant murine IL-2 (2 ng/ml) and IL-7 (5 ng/ml), in addition to IL-18 (100 ng/ml). IL-33 and IL-22 (100 ng/ml) were included as positive and negative controls, respectively. Quantification of the proliferation marker Ki67 revealed that ILC2s proliferated in response to IL-33 and IL-18, but not IL-22 (Fig 3C-E). Analysing supernatants of stimulated ILC2s for cytokine production by ELISA showed that only IL-18 and IL-33 triggered secretion of IL-5, IL-9, and IL-13 (Fig 3F), while IL-4 was not detected.

## IL-18 promotes in vivo expansion of colonic ILC2s during CR infection

To determine whether IL-18 directly contributes to ILC2s expansion during CR infection, we injected CR-infected mice with IL-18 binding protein (IL-18 BP) intraperitoneally (IP) on 2 and 3 dpi (Fig 4A). IL-18 BP-treated mice showed no significant differences in faecal bacterial load or colonic shortening as compared to vehicle-treated controls (Fig 4B and 4C). However, treated mice exhibited a significantly reduced frequency and total number of colonic lamina propria ILC2s (Fig 4D). In addition, IL-18 BP treatment reduced the proportion of NK cells but did not affect the ILC1 or ILC3 populations (S2A Fig).

To further investigate the role of IL-18 in driving ILC2 expansion, we used $Il22^{-/-}$ mice, which show reduced IL-18 expression during CR infection [34]. We first confirmed reduced $Il18$ expression in $Il22^{-/-}$ compared to WT mice at 4dpi (S3A and S3B Fig). Moreover, $Il22^{-/-}$ mice showed no changes in number or proportion of ILC2s upon CR infection compared to uninfected controls (Fig 5A). Interestingly, in contrast to CR-infected WT mice, increased frequency of ILC1s was seen in CR-infected $Il22^{-/-}$ mice. Similarly to WT mice, NK cells expanded and the proportion of ILC3s remained unchanged in CR-infected $Il22^{-/-}$ mice (S3C Fig).

To test whether reduced IL-18 levels were directly responsible for impaired ILC2s expansion in $Il22^{-/-}$ mice, we injected recombinant IL-18 (rIL-18) IP into $Il22^{-/-}$ mice at 2 and 3 dpi (Fig 5B). rIL-18 treatment did not affect faecal bacterial shedding but restored ILC2s expansion in $Il22^{-/-}$ mice (Fig 5C and 5D). Together, these findings indicate that IL-18 acts as a key upstream regulator of ILC2s expansion in the colonic lamina propria following CR infection.

## Neutralisation of IL-13 reduces secretion of anti-inflammatory cytokines

As IL-13 has been shown to affect infection outcomes following CDI [27], we assessed the functional role of ILC2 during early CR infection by injecting anti-IL-13 IP on days 2 and 4 post-infection (Fig 6A). Anti-IL-13 treatment did not significantly alter faecal bacterial shedding compared to control mice (Fig 6B). Similarly, disease severity remained comparable between treated and control mice, as measured by colon shortening, increased colon weight-to-length ratio, and faecal MPO levels (Fig 6C-E). Notably, while the levels of the pro-inflammatory cytokines TNF and IFNγ were unaffected (S4A Fig), mice receiving anti-IL-13 exhibited significantly reduced levels of the anti-inflammatory cytokines IL-10 and IL-22 (Fig 6F). These results suggest that ILC2s are involved in maintaining an anti-inflammatory environment that could facilitate epithelial repair.

## Discussion

ILC3s are one of the first responders to gut perturbation and infection with extracellular enteric pathogens, while ILC2s are mainly associated with helminth infection [23,24]. More recently, ILC2s have been implicated in colonic CDI and gastric *H.*

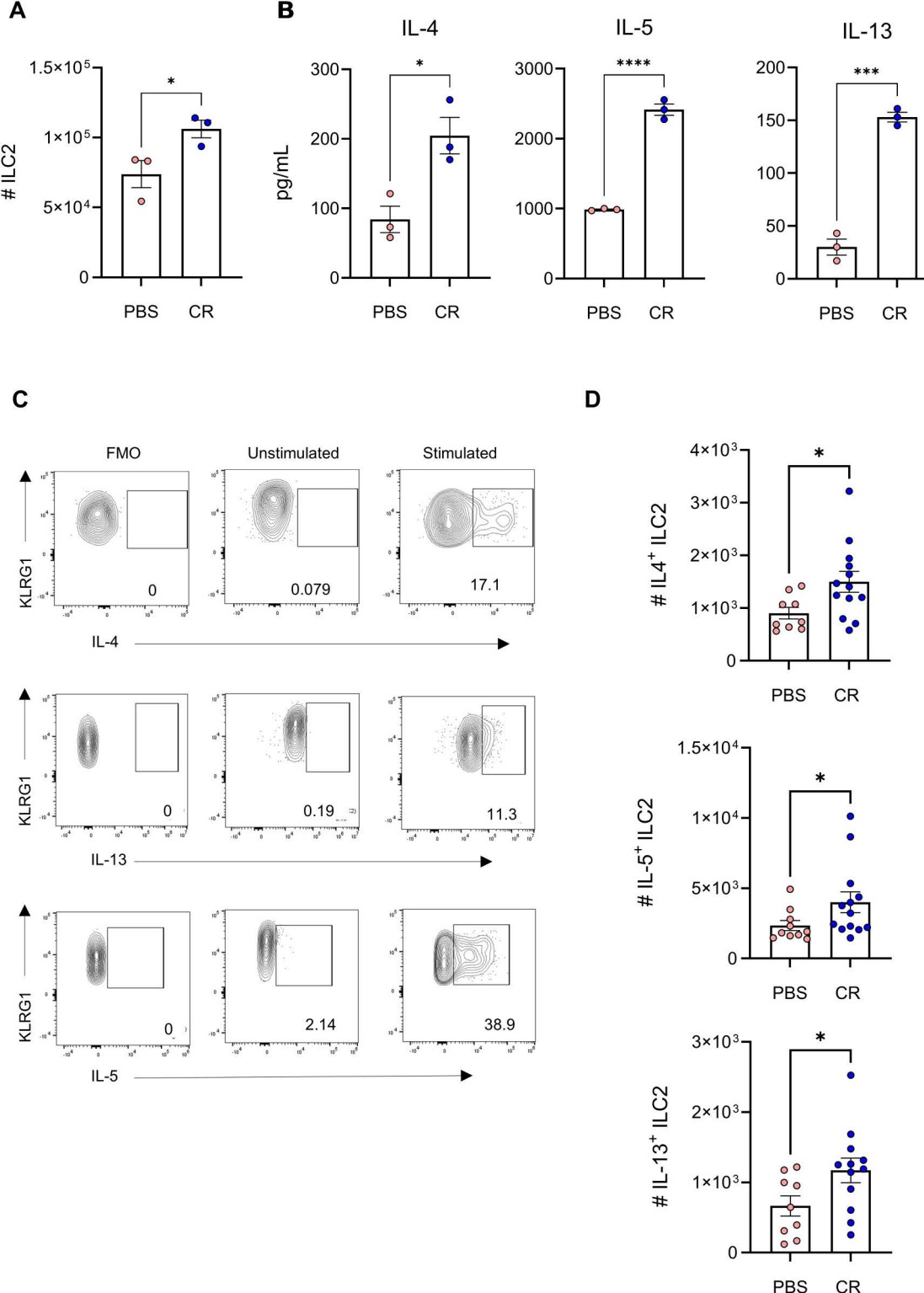

**Fig 2. ILC2 cytokine profile during CR infection.** A. Total ILC2s numbers sorted from pooled colons of uninfected and infected mice. B. IL-4, IL-5 and IL-13 levels in the supernatant from the same sorted cells measured by ELISA. C. Representative flow cytometry plots of cytokine expression from

fluorescent-minus-one (FMO), unstimulated and stimulated samples. D. Total number of IL-4+, IL-5+ and IL-13+ of sorted ILC2 cells from colons pooled from uninfected and infected mice. Cells were stimulated for 4 h in the presence of protein transport inhibitor cocktail to assess cytokine intracellularly by flow cytometry. Data shown are pooled values from three independent experiments. P values were determined on data plotted as mean ± SEM using Student's-t-test. *p < 0.05; ***p < 0.001; ****p < 0.0001.

*pylori* infection [25,26,28]. In this study we provide first evidence that colonic lamina propria ILC2s expand and produce type 2 cytokines in response to CR infection. While we do not yet know the exact role ILC2s, and their secreted cytokines, play during the infection, our data suggest that ILC2s are activated in the colonic lamina propria in response to Gram-negative bacterial pathogens.

ILC3s are known to be activated at 4 dpi, when CR sparsely and sporadically colonises the apex of the colonic crypts [16]. Here we show that at 4 dpi, the colonic tissue shows morphological signs of inflammation, i.e. shortening of the colon and secretion of MPO and IL-18. This is accompanied by a significant increase in population of colonic lamina propria ILC2s. Of note, of the major stimulators of ILC2s, we did not detect induction of *Il25* expression, while *TLSP* was undetected in the large intestine. In contrast, expression of *Il33* and *Il18* was induced in CR-infected mice. In terms of type 2 cytokines, stimulation of ILC2s isolated from infected mice, enhanced the production of IL-4, IL-5, and IL-13. Conversely, ILC2s isolated from the colons of naïve mice and maintained in vitro for 3 days and stimulated with IL-33 and IL-18 resulted in secretion of IL-5, IL-9, and IL-13; IL-4 was not detected. This shows important differences in ILC2 responses to IL-18 and IL-33 stimulation dependent on whether they were isolated from infected or naïve mice.

While their role is not yet known, it is possible that IL-4, IL-5, and IL-13 participate in mucus secretion and epithelial repair in early responses to CR colonisation in order to maintain barrier function. Mice deficient in IL-13 exhibit impaired barrier function and increased susceptibility to intestinal pathogens [36–38]. IL-5 is associated with the recruitment of eosinophils but can also enhance IgA production controlling bacterial spread in *H. pylori* infected mice [28,39,40]. IL-4, however, plays a role in regulating IgE responses in B cells and the differentiation of alternatively activated M2 macrophages [41]. In addition, IL-4, secreted from CD4+ T cells, has been shown to increase mucus production in CR-infected mice [42]. The significance of ILC2s activation during CR infection could lie in production of IL-4 and IL-13 to drive mucus production, in addition to IL-5 which can act synergistically with IL-4 to influence or enhance an antibody response, promoting bacterial clearance [43]. Moreover, ILC2s have been shown to synthesise and release acetylcholine (ACh) during parasitic nematode infection [44]. Depleting ACh production in T cells led to a higher bacterial burden and increased levels of IL-1β, IL-6 and TNFα in the gut of CR infected mice [45]. Therefore, ILC2s may play a supportive role by limiting tissue damage and promoting epithelial repair rather than directly controlling bacterial growth through type 2 cytokine secretion. This dual role of ILC2s in inflammation and repair is supported by recent work highlighting their plasticity and responsiveness to diverse cytokine milieus [46].

In addition to promoting goblet cell responses during helminth infection [47], ILC2s derived IL-13 may also play a supportive role in shaping the mucosal cytokine environment during bacterial infection. In our model, neutralisation of IL-13 during early CR infection did not significantly alter clinical disease parameters but resulted in reduced colonic levels of IL-10 and IL-22. While IL-10 is an important regulator in preventing pro-inflammatory responses [48], *Il10-/-* mice display faster pathogen clearance and attenuated colitis post CR infection [49]. However, IL-22 has a protective role during CR infection and *Il22-/-* mice are susceptible to CR [18,35]. These results indicate a broader immunomodulatory function for ILC2s, beyond type 2 effector responses, and suggest their involvement in maintaining an anti-inflammatory milieu that could facilitate epithelial repair.

We show that IL-18 and IL-33, detected at 4 dpi, stimulate colonic ILC2s proliferation and cytokine production in vitro. Treating mice with IL-18 BP at days 2 and 3 post-CR infection inhibited ILC2s expansion, and treating *Il22-/-* mice with rIL-18 rescued ILC2s expansion, together providing direct evidence that IL-18 plays a key role in ILC2s expansion during infection. While IL-33 is a well-established ILC2s activator [50,51], our findings add to the emerging evidence that IL-18,

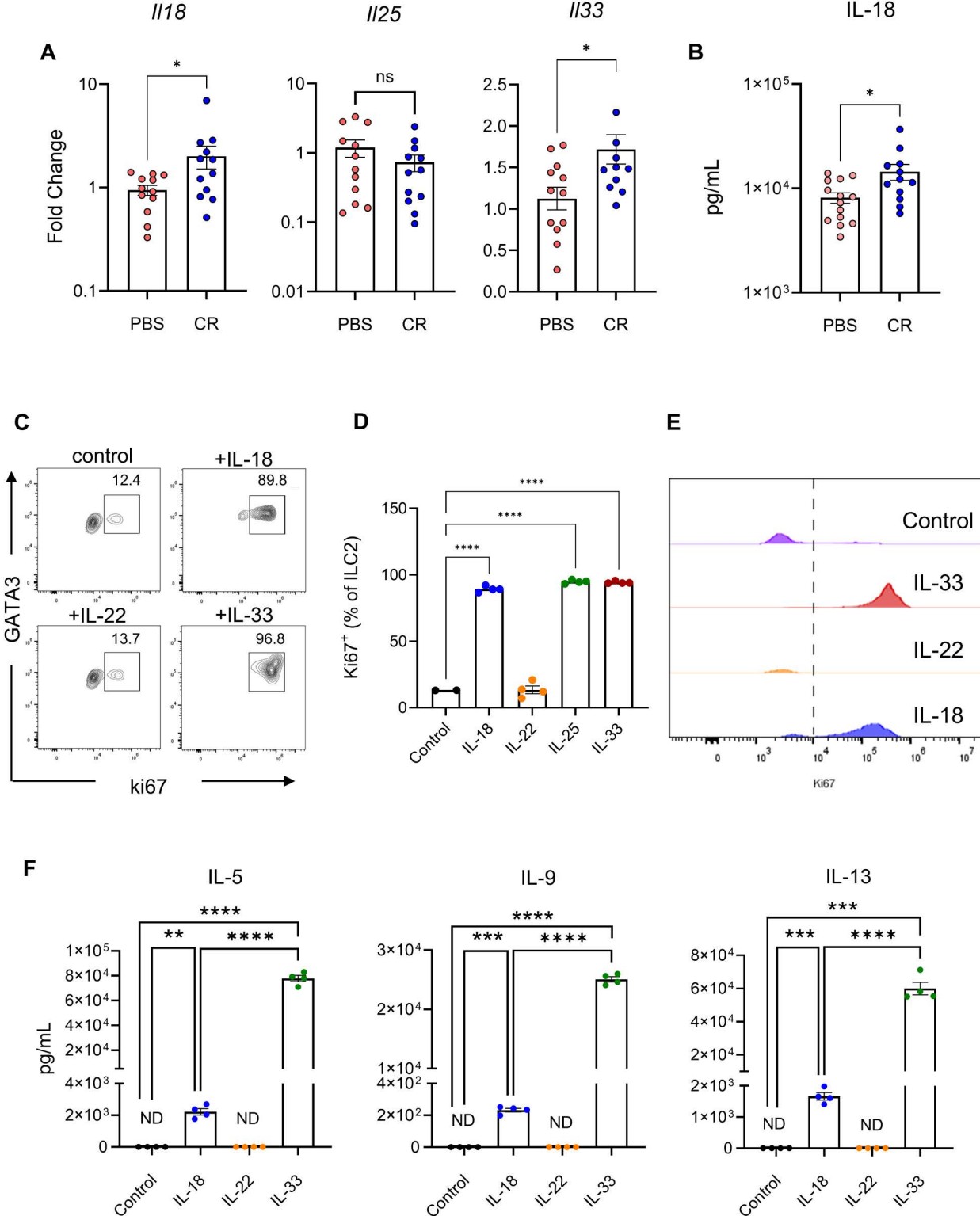

**Fig 3. Colonic lamina propria ILC2s proliferate in response to IL-18.** A. RT-qPCR analysis of *Il18*, *Il25* and *Il33* transcript expression in colons of CR-infected and PBS-dosed mice at 4 dpi. Each data point represents one mouse. B. IL-18 levels secreted from explant colonic tissue and detected by

ELISA. C. Representative flow cytometry plots and D. Summary data for Ki67 expression (represented as percentage of ILC2) by FACS sorted colonic lamina propria ILC2s. Cells were cultured for 3 days in vitro with recombinant IL-2 and IL-7 (control) in addition to IL-18, IL-22 or IL-33. E. Histogram showing positive and negative peaks of proliferative ILC2s in response to different cytokines. F. IL-5, IL-9 and IL-13 levels in the supernatant from cultured ILC2s measured by ELISA. Cells pooled from n = 10 mice per experiment. Data shown are pooled values from four independent experiments. P values were determined on data plotted as mean ± SEM using Student's-t-test (A-B) and ANOVA with Bonferroni post-test for multiple comparisons (D, F). ns: non-significant; *p < 0.05; **p < 0.01; ***p < 0.001; ****p < 0.0001.

traditionally associated with type 1 responses, can activate ILC2s under certain conditions. Skin ILC2 subsets have been shown to be activated dominantly by IL-18 via the IL-18R1 receptor, which is highly expressed in skin ILC2s and can be detected on lung and bone marrow, but not on small intestine ILC2s [30]. Our data suggest that the IL-18R1 receptor may also be expressed on ILC2s in the large intestine.

Recent studies have highlighted the critical role of IL-18 in modulating ILC3s function during CR infection. IL-18 was shown to specifically trigger the production of ILC3-derived IL-22 via HIF-1α activation, contributing to mucosal protection and host survival [52]. Our findings add a new dimension to this pathway by demonstrating that IL-18 also regulates ILC2s expansion in the early phase of CR infection. This suggests that IL-18 may coordinate the activity of both ILC2s and ILC3s to orchestrate epithelial protection. The downstream impact of ILC2-derived IL-13 on IL-22 production further indicates potential crosstalk between these two innate lymphoid subsets, either directly or via modulation of the epithelial niche.

While ILC3s are traditionally associated with antimicrobial responses and barrier defence [15], our data suggest a supportive role for ILC2s in shaping the cytokine environment, possibly by promoting tissue repair and limiting collateral damage. Such coordination may be especially relevant in non-invasive infections like CR and CDI, where epithelial preservation is paramount. It is plausible that ILC2s act to mitigate epithelial damage while ILC3s focus on microbial containment. These findings warrant further investigation into the temporal and spatial coordination between ILC2s and ILC3s in mucosal immune responses, and how shared cytokine signals such as IL-18 may integrate these functions during infection and repair.

Taken together, this study shows that colonic ILC2s proliferate and secrete type 2 cytokines in response to CR infection, driven, at least in part, by IL-18. Further studies using genetically modified mice are needed to determine the functional role of ILC2s in host responses to CR and how widespread this response is in other bacterial enteric infections.

## Materials and methods

### Mice and ethics statement

Female C57BL/6 mice of 6–10 weeks age were purchased from Charles River UK. The *Il22*<sup>-/-</sup> mice were maintained in C576BL/6 background in homozygous condition and were housed and bred in dedicated animal facilities of Imperial College London. All mice were maintained at the Central Biomedical Services (CBS) facility at Imperial in pathogen-free conditions at 20–22 °C and 30–40% humidity on 12 h light/dark cycle.

### Ethics statement

All animal procedures were conducted in accordance with Animals Scientific Procedures Act 1986 and U.K. Home Office guidelines and reviewed by the Imperial College Animal Welfare Ethical Review Body (AWERB). The in vivo procedures were approved by a home office licence, number PP7392693.

### CR infection

CR ICC169 cultures were prepared in 15 ml LB supplemented with 50 µg/µL nalidixic acid and grown at 37 °C overnight. The cultures were centrifuged, resuspended with 1.5 ml phosphate-buffered saline (PBS) and each mouse received 200 µl (approximately $10^9$ CFU) by oral gavage [13]. Uninfected mice (mock infected) received 200 µl of PBS. To quantify

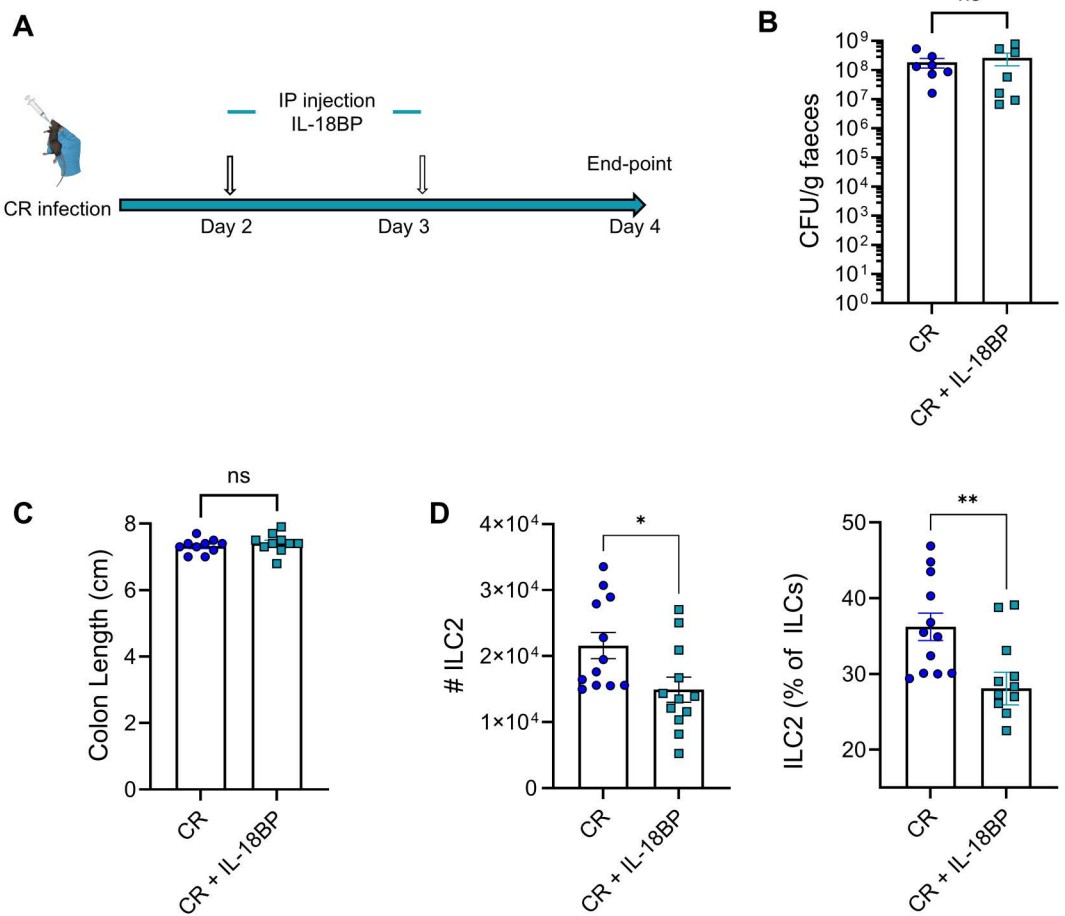

**Fig 4. Injection of IL-18 BP into CR-infected mice results in lower ILC2s expansion.** A. Experimental plan. B. Faecal CR shedding at 4 dpi. C. Colon length at 4 dpi. D. Frequency and absolute number of ILC2s is reduced in the colon of CR-infected mice treated with IL-18 BP. Data shown are pooled values from three independent experiments. P values were determined on data plotted as mean ± SEM using Student's-t-test. ns: non-significant; *p < 0.05; **p < 0.01.

the CFU per gram of faeces, faecal samples were homogenised, diluted and plated onto agar plates and viable bacteria were counted as previously described [13]. Infected mice that did not reach the threshold of 1 x $10^7$ CFU/g of faeces were excluded from further experimental processing and analysis.

## Administration of IL-18 BP, rIL-18 and anti-IL-13

Recombinant human IL-18 binding protein (rhIL-18 BP, Cat No. 119-BP, R&D Systems), recombinant mouse IL-18 (rIL-18, Cat No. 9139-IL, R&D Systems) or anti-mouse IL-13 antibody (Cat No. 16-7135-81, Thermo Fischer Scientific) were freshly prepared in PBS + 0.1% normal mouse serum and given to mice by intraperitoneal injection, in a volume of 0.1 ml on days 2 and 3 p.i. for IL-18 BP and rIL-18 experiments, and on 2 and 4 dpi for anti-IL-13 experiments at a dose of 5 mg/kg, 1 mg/kg and 150 μg/mouse respectively. Control groups received 0.1 mL of PBS + 0.1% normal mouse serum.

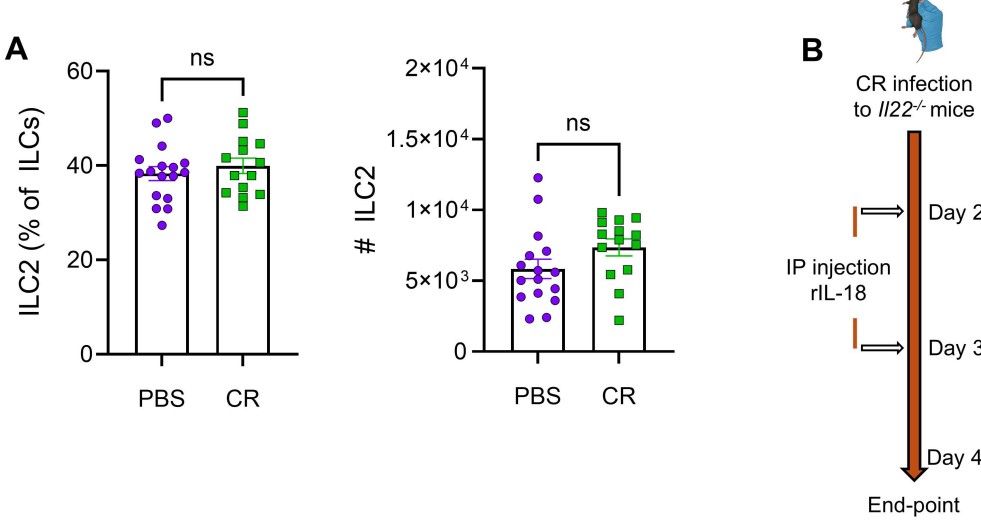

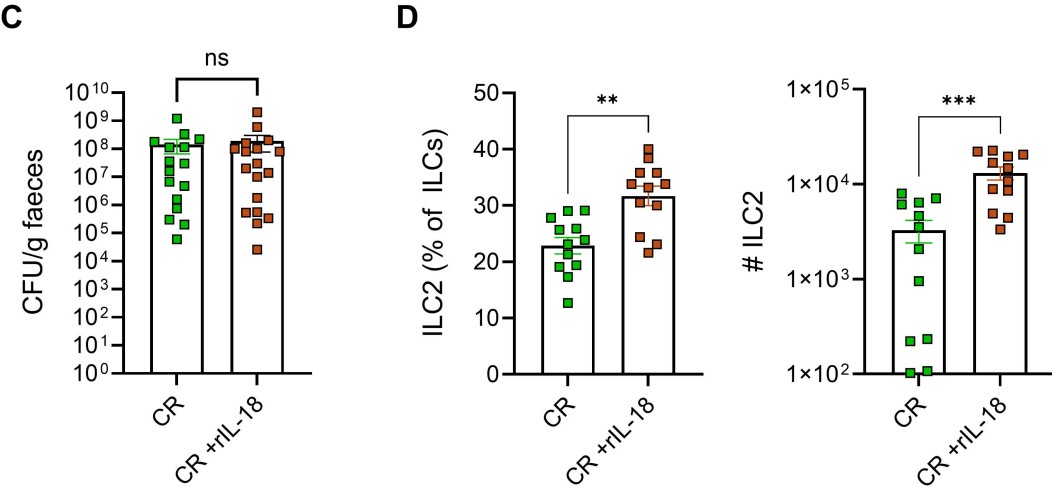

**Fig 5. *Il22⁻′⁻* mice display lower IL-18-mediated ILC2s expansion.** A. Frequency and absolute number of ILC2s in CR infected *Il22⁻′⁻* mice at 4 dpi. B. Experimental plan of rIL18 treatment. C. Faecal CR shedding at 4 dpi. D. Frequency and absolute number of ILC2s post rIL-18 treatment. Data shown are pooled values from three independent experiments. P values were determined on data plotted as mean±SEM using Student's-t-test. ns: non-significant; **p<0.01; ***p<0.001.

## Cytokines and antibodies

Recombinant murine cytokines were purchased from PeproTech (IL-2, IL-7, IL-22, and IL-33) or R&D Systems (IL-18 and TSLP). All cytokines were lyophilised in PBS and stored at -80 °C. The following antibodies were purchased from Biolegend or BD Biosciences and used for flow cytometry: Fc Block (anti-mouse CD16/CD32 mAb, clone 2.4G2, BD Biosciences), CD3 (145- 2C11), B220 (RA3-6B2), CD19 (6D5), TER119 (TER-119), Gr-1 (RB6-8C5), CD5 (53-7.3), FcεRI (MarI), CD11c (N418), F4/80 (BM8), CD45 (30F11), CD90.2 (30-H12), Nkp46 (29A1.4), CD4 (RM4–5), CD127 (A7R34), KLRG1 (2F1), NK1.1 (PK136), GATA3 (L50-823), RORγt (Q31-378), IL-4 (11B11), IL-5 (TRFR5), IL-13 (ebio13A) and Ki67 (SolA15).

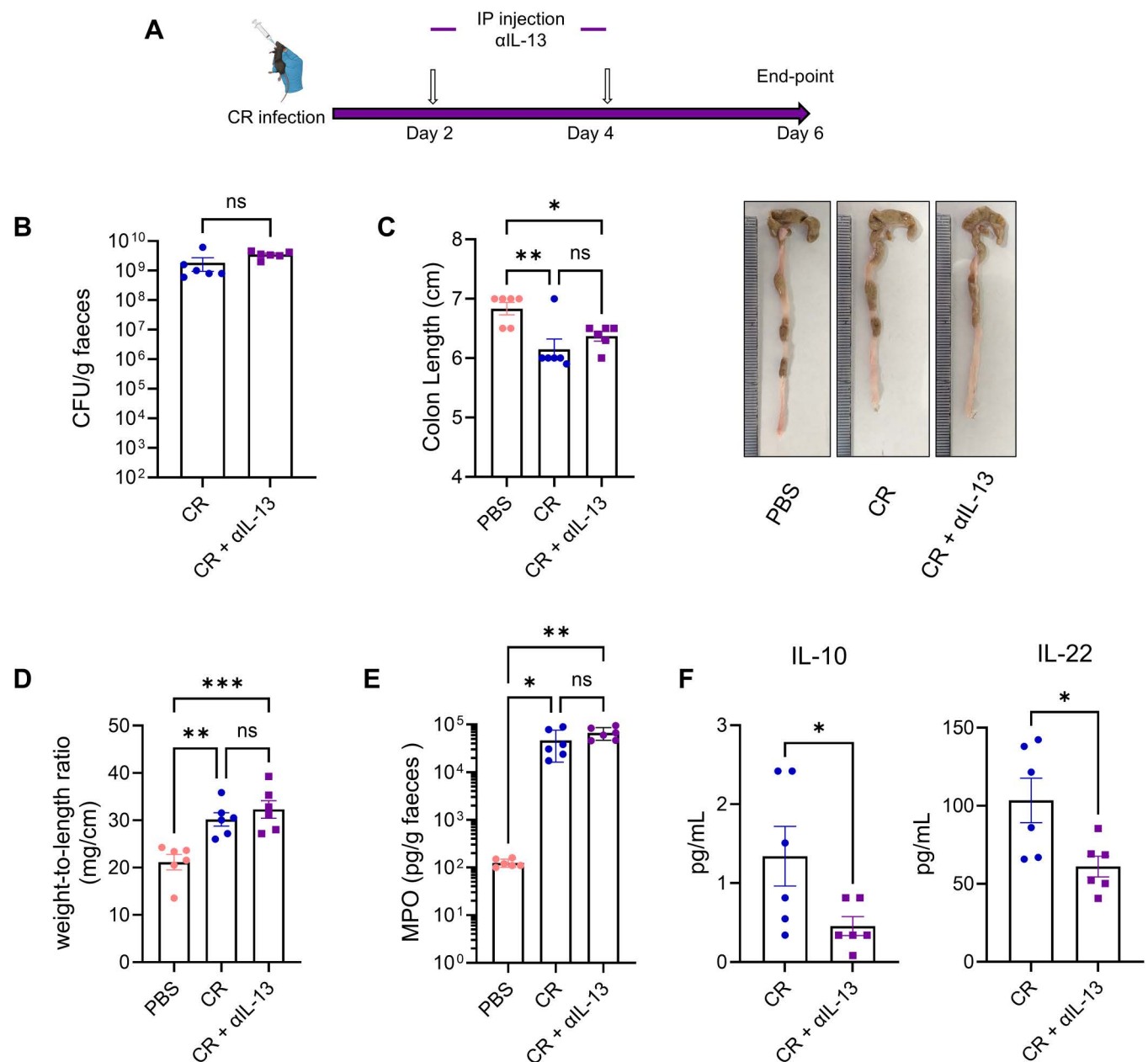

**Fig 6. Neutralisation of IL-13 post CR infection results in lower anti-inflammatory cytokine levels.** A. Experimental plan. B. Faecal CR shedding at 4 dpi. C. Representative and quantitative colon length. D. Weight-to-length ration of colon at 6 dpi. E. Faecal MPO levels measured by ELISA. F. Levels of IL-10 and IL-22 in colonic explant culture. Data shown are pooled values from two independent experiments with three mice in each group per experiment. P values were determined on data plotted as mean±SEM using Student's-t-test (B, F) and ANOVA with Bonferroni post-test for multiple comparisons (C-E). ns: non-significant; *p < 0.05; **p < 0.01; ***p < 0.001.

### Isolation of lamina propria immune cells and IECs from mouse colons

Lamina propria cells were isolated from mouse colons as previously described [53]. Briefly, cleaned colons were cut longitudinally then incubated for 20 min at 37 °C shaking in Hanks' balanced salt solution (HBSS; Ca2+ and Mg2+ free) supplemented with 2% FBS, 10 mM EDTA and 1 mM DTT. Residual tissues were digested at 37°C for ~40–50 min in RPMI 1640

containing 62.5 µg/mL Liberase, 50 µg/mL DNase I and 2% FBS. Mononuclear cells were isolated with 40–80% Percoll gradient and washed twice.

## Flow cytometry and cell sorting

For the analysis of cytokine expression, isolated cells from 15 colons were pooled, equally split and were stimulated with 1x cell stimulation cocktail (Cat No. 00–4970, eBioscience) for 4 h at 37°C/ 5% CO2 in the presence or absence of 1x protein transport inhibitor cocktail (Cat No. 00–4980, eBioscience).

To stain for extracellular markers, single-cell suspensions or stimulated cells were plated in a 96 well V bottomed plate and stained for 10 min with LIVE/DEAD fixable blue (Thermo Fisher, diluted in D-PBS) to detect and exclude dead cells from subsequent analysis. Cells were then treated for 20 min with Fcγ receptor block followed by surface marker staining using fluorophore-conjugated monoclonal antibodies (BioLegend, BD Biosciences or eBioscience). All incubations were performed at 4°C in the dark, unless otherwise stated. Negative (unstained and live/dead) along with fluorescent-minus-one (FMO) controls were considered to estimate background fluorescence.

For intracellular staining, cells were first fixed for 20–30 min at room temperature in the dark, using the eBioscience Forkhead box protein 3 (Foxp3)/transcription factor fixation buffer set to stain for transcription factors or the IC fixation buffer for the detection of intracellular cytokines. Fixed cells were then stained for 1 h with intracellular antibody cocktail diluted in permeabilisation buffer. Cells were then washed prior to analysis on an Aurora flow cytometer (Cytek Biosciences).

For ILC2 sorting, lineage negative population (CD3ε, CD4, CD8α, CD5, CD19, CD11c, Gr1, F4/80, FcεRIa, NK1.1, CD11b, and TER119) were excluded by Magnetic Cell sorting system (MACS, Miltenyi Biotec) from single-cell suspension. MACS separated cells were blocked with anti-CD16/CD32 for 20 min on ice then stained with CD45, CD127, KLRG1, CD90.2 and lineage cocktail antibodies for 1 hour on ice. After staining, cells were washed and resuspended in DAPI 5 min prior to sorting and ILC2 cells were sorted on a FACSAria III cell sorter (BD Biosciences). The gating strategy is described in S5 Fig.

## ILC2 in vitro culture and proliferation assay

For each experimental repeat, colons from 15 mice were pooled and sorted as a single sample, and isolated ILC2 cells were split equally between different conditions (ca. 20,000 cells per condition). ILC2s were cultured for 3 days at 37°C/ 5% CO2 in U-bottom 96-well plate in complete RPMI medium containing recombinant murine IL-2 (2 ng/ml), IL-7 (5 ng/ml) and either IL-18, IL-22 or IL-33 (100 ng/ml). Cells were re-stained using the extracellular ILC2 marker panel followed by intranuclear Ki67 staining, and cell proliferation was assessed by flow cytometry.

## RNA isolation, reverse transcription and quantitative real-time PCR

Total RNA was extracted from colonic tissue using the RNeasy Mini Kit (QIAGEN) and converted to cDNA using iScript cDNA synthesis kit (BIO-RAD) in accordance with the manufacturer's instructions. The RT-qPCR reactions were carried out using the PowerUP SYBR green PCR master mix (Applied Biosystems) and run in duplicates in a 7500 Fast RT- PCR thermocycler (Applied Biosystems). Relative gene expression of Il18, Il25, Il33 and Tslp was calculated by the comparative cycle threshold (Ct) method $2^{-\Delta\Delta Ct}$, using Hprt, Actb and Gapdh as housekeeping genes. Primer sequences were as follows: Il18 forward: GACTCTTGCGTCAACTTCAAGG; Il18 reverse: CAGGCTGTCTTTTGTCAACGA; Il25 forward: ACAGGGACTTGAATCGGGTC; Il25 reverse: TGGTAAAGTGGGACGGAGTTG; Il33 forward: TCCAACTCCAAGATTTC-CCCG; Il33 reverse: CATGCAGTAGACATGGCAGAA; Tslp forward: ACGGATGGGGCTAACTTACAA and Tslp reverse: AGTCCTCGATTTGCTCGAACT.

## Cytokine analysis by ELISA

The levels of IL- 4, IL-5, IL-9, IL-13, and IL-18 were determined using Duoset ELISA kits (R&D Systems) and levels of IL-10, IL-22, TNF and IFNγ were determined using LEGENDplex assay kit (BioLegend, catalogue number 741044) following the

manufacturer's protocol. For tissue explant, 1 cm colon fragments were transferred to a 96-well plate containing RPMI buffer supplemented with 10% faetal bovine serum (FBS), 100 U/ml penicillin, 100 µg/ml streptomycin and incubated in a 37°C humidified 5% CO2 incubator for 24 h. Supernatants from cultured ILC2 or colonic explants were collected, and cytokines were analysed by ELISA (R&D Systems) or LEGENDplex assay kit according to the manufacturer's instructions.

### Statistical analysis

Flow cytometry data were analysed using FLowJo software (Tree Star). Graphs and statistical tests were carried out using GraphPad Prism 10.4 software and data were expressed as mean ± SEM. Statistical differences were calculated using the unpaired Student's t test for two groups and ANOVA for multiple comparisons. One sample Students t-test was performed when comparing a group to other group where no values were determined (ND). A p value of <0.05 is considered significant, $*p < 0.05$, $**p < 0.01$, $***p < 0.001$, $****p < 0.0001$.

### Image generation

The Images depicting the experimental strategy in 4A, 5B and 6A are drawn by the author using Microsoft PowerPoint. However, the cartoon of mouse being oral gavaged in all three images was generated using **OpenAI's DALL·E (GPT-4o) model** via the ChatGPT platform. The image was created using the following prompt:

"Semi-realistic medical artwork showing a laboratory mouse being fed a liquid through a gavage needle. The mouse, with dark fur and striking features, is held securely by a gloved hand while a syringe administers the liquid into its mouth; the careful detailing of the mouse's body, the gloved hand, and the syringe creates a clear, focused depiction against a simple white backdrop."

All images created through this platform are free for commercial use, redistribution, and publication under an irrevocable license.

### Supporting information

**S1 Fig. Proportion of ILCs post CR infection.** A. Frequency of ILC1, ILC2, ILC3 and NK cells, B. Levels of IL-22 in colonic explant culture and C. Faecal lipocalin levels post CR infection at 4 dpi. Data shown are pooled values from two independent experiments. P values were determined on data plotted as mean ± SEM using ANOVA with Bonferroni post-test for multiple comparisons (A) and Student's-t-test (B-C). ns: nonsignificant; $**p < 0.01$; $***p < 0.001$; $****p < 0.0001$. (TIF)

**S2 Fig. Proportion of ILCs post IL-18 BP treatment.** A. Frequency of ILC1 and B. NK cells post IL-18 BP treatment to CR infected mice. Data shown are pooled values from three independent experiments. P values were determined on data plotted as mean ± SEM using ANOVA with Bonferroni post-test for multiple comparisons. ns: nonsignificant; $*p < 0.05$; $***p < 0.001$. (TIF)

**S3 Fig. CR infection to Il22-/- mice.** A. RT-qPCR for Il18 from colonic tissue from CR infected mice at 4 dpi. B. Faecal CR shedding. C. Proportion of ILC1, ILC3 and NK cells in Il22-/- mice post CR infection at 4 dpi. Data shown are pooled values from three independent experiments. P values were determined on data plotted as mean ± SEM using Student's-t-test (A) and ANOVA with Bonferroni post-test for multiple comparisons (C). ns: non-significant; $*p < 0.05$; $**p < 0.01$; $****p < 0.00$. (TIF)

**S4 Fig. Cytokine levels in colonic explant culture.** A. Levels of TNF and B. IFNγ in colonic explant of mice treated with anti-IL-13. Data shown are pooled values from two independent experiments with three mice in

each group per experiment. P values were determined on data plotted as mean ± SEM using Student's-t-test. ns: non-significant.
(TIF)

**S5 Fig. Gating strategy for sorting ILCs from colonic lamina propria.**
(TIF)

## Acknowledgments

We thank Jessica Rowley and Larissa Zarate Garcia for their technical assistance with flow cytometry and cell sorting.

## Author contributions

**Conceptualization:** Rita Berkachy, Gad Frankel.

**Data curation:** Rita Berkachy.

**Formal analysis:** Rita Berkachy, Vishwas Mishra, Priyanka Biswas.

**Funding acquisition:** Gad Frankel.

**Investigation:** Rita Berkachy, Vishwas Mishra.

**Methodology:** Rita Berkachy, Vishwas Mishra.

**Writing – original draft:** Rita Berkachy, Gad Frankel.

**Writing – review & editing:** Vishwas Mishra, Gad Frankel.

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
