## [Decision Letter · Decision Letter 0]

Citrobacter rodentium infection activates colonic lamina propria group 2 innate lymphoid cells

PLOS Pathogens

Dear Dr. Frankel,

Thank you for submitting your manuscript to PLOS Pathogens. Your study demonstrated that Citrobacter rodentium (CR) infection in mice induced group 2 innate lymphoid cells (ILC2s), which was further validated in vitro culture assays. However, the role of ILC2s in CR infection remains unknown, which should be addressed by performing functional assays as requested by reviewer 1. We feel that your manuscript has merit but does not fully meet PLOS Pathogens's publication criteria as it currently stands. Therefore, we invite you to submit a revised version of the manuscript that addresses the above concern and the other points raised during the review process.

Please submit your revised manuscript within 60 days. If you will need more time than this to complete your revisions, please reply to this message or contact the journal office at plospathogens@plos.org. Please include the following items when submitting your revised manuscript:

We look forward to receiving your revised manuscript.

Kind regards,

Guangming Zhong

Academic Editor

PLOS Pathogens

David Skurnik

Section Editor

Editor-in-Chief

PLOS Pathogens

orcid.org/0000-0003-2946-9497

Editor-in-Chief

PLOS Pathogens

orcid.org/0000-0002-7699-2064

**Journal Requirements:**

At this stage, the following Authors/Authors require contributions: Gad Frankel. Please ensure that the full contributions of each author are acknowledged in the "Add/Edit/Remove Authors" section of our submission form.

Potential Copyright Issues:

- Please confirm that you are the photographer of Figure 1D, or provide written permission from the photographer to publish the photo(s) under our CC BY 4.0 license.

- Figure 4A. Please confirm whether you drew the images / clip-art within the figure panels by hand. If you did not draw the images, please provide a link to the source of the images or icons and their license / terms of use; or written permission from the copyright holder to publish the images or icons under our CC BY 4.0 license. Alternatively, you may replace the images with open source alternatives. See these open source resources you may use to replace images / clip-art:

5) We note that your Data Availability Statement is currently as follows: "All the data generated in this study are included within the submitted paper and supplementary material". Please confirm at this time whether or not your submission contains all raw data required to replicate the results of your study. Authors must share the “minimal data set” for their submission. PLOS defines the minimal data set to consist of the data required to replicate all study findings reported in the article, as well as related metadata and methods (https://journals.plos.org/plosone/s/data-availability#loc-minimal-data-set-definition).

**Reviewers' Comments:**

Reviewer's Responses to Questions

**Part I - Summary**

Reviewer #1: Authors study the role of ILC2s in Citrobacter rodentium infection. They show that C. rodentium infection of WT mice leads to increased numbers of ILC2s in lamina propria. Purified ILC2s from C. rodentium infected mice produced more IL-4, IL-5, IL-13 compared to control ILC2s from naïve mice. ILC2s purified from naïve mice also produced IL-5, IL-9, IL-13 and proliferated when stimulated in vitro with IL-18 and IL-33. IL-18BP treated WT mice had lower number of ILC2s in lamina propria. Although, there are interesting observations of ILC2s responding to IL-18, similarly to skin ILC2s in skin, the provided data do not provide conclusive evidence for the role of ILC2s in protection or pathology in response to C. rodentium.

Reviewer #2: In this manuscript, the Frankel group reported the activation of group 2 innate lymphoid cells (ILC2s) during Citrobacter rodentium (CR) infection in mice. Traditionally, ILC2s are associated with immune responses to helminth infection and allergic inflammation and ILC3s are more important in protecting against infection with the enteric mouse pathogen including CR. The authors showed that colonic lamina propria ILC2s proliferate in response to CR infection and secrete type 2 cytokines IL-4, IL-5 and IL-13. Moreover, naïve colonic ILC2s, following in vitro stimulation with IL-18, secret type 2 cytokines, and injection of IL-18 binding protein (IL18BP) post CR infection blocks the activation of ILC2s. Although the precise role of ILC2s in CR infection is not known yet, the authors proposed that ILC2s are activated in response to CR infection, where stimulation with IL-18 plays a role in inducing proliferation and secretion of type 2 cytokines. The experiments are well controlled and support the claims by large and the study is of potential interest to the field. FW

**Part II – Major Issues: Key Experiments Required for Acceptance**

Reviewer #1: Authors need to address the major issue: perform loss of function and gain of function ILC2 targeting experiments to prove the role of ILC2s in protection against C. rodentium, independently from ILC3s.

Comments to Figures:

Figure 1.

- Authors should compare ILC2s to ILC1s and ICL3s to show if there is preferential expansion of ILC2s after infection.

- Authors need to provide KI67 (BrdU) ILC2s data to support their conclusion of ILCs proliferation in vivo after C. rodentium infection.

Figure 2.

-Not described which day after infection mice were analyzed

- Fig 2A shows average 1X10^5 ILC2s, whereas Fig 1H shows average 1.5 X10^4. Not clear how many mice were combined in Fig 2.

- Panel D lacks statistical evaluation

- No FACS plots shown for IL-4, IL-13.

Figure 3.

Panel A. Description says these are enriched IEC, but RNA is prepared from total tissue. Whether these are purified IECs or total tissue?

Panel C-F. Not specified how many ILC2s were analyzed.

Figure 4.

Authors perform IL-18 treatment in B6 and IL-22KO but show only ILC2s in WT mice. What are results in IL-22KO? Whether IL-18 mediated stimulation of ILC2s is sufficient to rescue intestinal pathology in IL-22 KO mice?

Reviewer #2: 1. Figure 4A, the panel shows C57Bl/6 WT or IL-22 KO mouse; the figure legend title wrote “IL-22 KO mice”; while the figure legend Figure 4B reads “CR infection C57BL/6J mice” – please double check which strain(s) was used in the experiment treating with IL-18BP.

2. The experimental evidence that supports the claimed protective effect of ILC2s is missing. For the CR-infected and IL-18BP injected mice at 4 dpi, the authors could better characterize the histology of the distal colons, measure their colon lengths, and assess the fecal MPO levels by ELISA, as did in the Figure 1C-1F.

3. IL-18 is also known to play a critical role for ILC3s function in CR-induced colitis (PMID: 38019650), while the current study suggests the importance of IL-18 in ILC2s function – how the activation of ILC3s and ILC2s are coordinated during CR infection? CR is non-invasive whereas Clostridium difficile and Helicobacter pylori are invasive; the speculated supportive role of ILC2s by limiting tissue damage and promoting epithelial repair could be the same or distinct under during these different infections? More in-depth discussion should be included.

**Part III – Minor Issues: Editorial and Data Presentation Modifications**

Reviewer #1: Dose of C. rodentium used in experiments is not specified

Reviewer #2: Figure 2C, remove “IL-5” in the flow cytometry plots, as X-axis showed it already. Increase the size of the numeric percentage of the gated cells. The same concerns for the Figure 3C.

Figure 3A, add “ns” to the panel of Il25, even not significant statistically.

Figure 2D and Figure 3F, statistical analyses should be included.

PLOS authors have the option to publish the peer review history of their article (what does this mean? ). If published, this will include your full peer review and any attached files.

**Do you want your identity to be public for this peer review?** For information about this choice, including consent withdrawal, please see our Privacy Policy .

Reviewer #1: No

Reviewer #2: No

**Figure resubmission:**

**Reproducibility:**



---

## [Editor Report · Decision Letter 1]

Dear Dr. Frankel,

We are pleased to inform you that your manuscript 'Citrobacter rodentium infection activates colonic lamina propria group 2 innate lymphoid cells' has been provisionally accepted for publication in PLOS Pathogens.

Best regards,

Guangming Zhong

Academic Editor

PLOS Pathogens

David Skurnik

Section Editor

PLOS Pathogens

Sumita Bhaduri-McIntosh

Editor-in-Chief

PLOS Pathogens

orcid.org/0000-0003-2946-9497

Michael Malim

Editor-in-Chief

PLOS Pathogens

orcid.org/0000-0002-7699-2064
---

## [Editor Report · Acceptance letter]

Dear Prof. Frankel,

We are delighted to inform you that your manuscript, "*Citrobacter rodentium*  infection activates colonic lamina propria group 2 innate lymphoid cells," has been formally accepted for publication in PLOS Pathogens.

Best regards,

Sumita Bhaduri-McIntosh

Editor-in-Chief

PLOS Pathogens

orcid.org/0000-0003-2946-9497

Michael Malim

Editor-in-Chief

PLOS Pathogens

orcid.org/0000-0002-7699-2064